# LECTURE

# Bayliss Starling Prize Lecture 2023: Neuropeptide-Y being 'unsympathetic' to the broken hearted

Benjamin Bussmann , Thamali Ayagama, Kun Liu, Dan Li  and Neil Herring 

*Burdon Sanderson Cardiac Science Centre, Department of Physiology, Anatomy and Genetics, University of Oxford, Oxford, UK*

Handling Editors: Harold Schultz & Kalyanam Shivkumar

The peer review history is available in the Supporting Information section of this article (https://doi.org/10.1113/JP285370#support-information-section).

**Abstract**  William Bayliss and Ernest Starling are not only famous as pioneers in cardiovascular physiology, but also responsible for the discovery of the first hormone (from the Greek 'excite or arouse'), the intestinal signalling molecule and neuropeptide secretin in 1902. Our research group focuses on neuropeptides and neuromodulators that influence cardiovascular autonomic control as potential biomarkers in disease and tractable targets for therapeutic intervention. Acute myocardial infarction (AMI) and chronic heart failure (CHF) result in high levels of cardiac sympathetic stimulation, which is a poor prognostic indicator. Although beta-blockers improve mortality in these conditions by preventing the action of the neurotransmitter noradrenaline, a substantial residual risk remains. Recently, we have identified the sympathetic co-transmitter neuropeptide-Y (NPY) as being released during AMI, leading to larger infarcts and life-threatening arrhythmia in both animal models and patients. Here, we discuss recently published data demonstrating that peripheral venous NPY levels are associated

**Neil Herring** completed his DPhil and medical degree at Oxford University before training in Cardiology. He was awarded Intermediate and Senior Fellowships from the British Heart Foundation and now leads a translational research group alongside clinical work as a Professor of Cardiovascular Medicine. **Ben Bussmann** is a Cardiology Registrar undertaking a BHF funded DPhil who obtained his medical degree from Cambridge and Oxford Universities. **Thamali Ayagama** is a postdoctoral researcher who received an MSc from the University of Bangalore, India, and DPhil from Oxford. **Kun (Kevin) Liu** obtained his medical degree from Huazhong University of Science and Technology, China, an MPhil from Cambridge University and DPhil from Oxford where he is now a postdoctoral researcher. **Dan Li** received her PhD from Chonbuk National University, South Korea before coming to the University of Oxford where she is now an Associate Professor (photograph left to right: Kevin, Dan, Neil, Thamali and Ben).

with heart failure hospitalisation and mortality after AMI, and all cause cardiovascular mortality in CHF, even when adjusting for known risk factors (including brain natriuretic peptide). We have investigated the mechanistic basis for these observations in human and rat stellate ganglia and cardiac tissue, manipulating NPY neurochemistry at the same time as using state-of-the-art imaging techniques, to establish the receptor pathways responsible for NPY signalling. We propose NPY as a new mechanistic biomarker in AMI and CHF patients and aim to determine whether specific NPY receptor blockers can prevent arrhythmia and attenuate the development of heart failure.

(Received 5 March 2024; accepted after revision 1 May 2024; first published online 7 June 2024)

**Corresponding author** N. Herring: Burdon Sanderson Cardiac Science Centre, Department of Physiology, Anatomy and Genetics, University of Oxford, Oxford OX1 3PT, UK.    Email: neil.herring@dpag.ox.ac.uk

**Abstract figure legend** The hallmark of cardiac disease is autonomic dysregulation, characterised by a state of chronic sympathoexcitation. Neuropeptide-Y (NPY) is a sympathetic co-transmitter that is released by sympathetic neurons and circulating venous levels are elevated in a range of cardiac disease, such as myocardial infarction and chronic heart failure. NPY has direct effects on cardiomyocytes, vascular smooth muscle cells and autonomic nerves through its Y receptors, resulting in adverse cardiac remodelling, pro-arrhythmic electrophysiological changes, vasoconstriction and parasympathetic inhibition. Large prospective cohort studies have demonstrated that these effects ultimately lead to adverse cardiac events and increased mortality in patients. An understanding of the role of co-transmitters such as NPY may ultimately lead to novel therapeutic targets and biomarkers to improve risk stratification and prognostication in patients with cardiac disease.

## Introduction

In 1902, Bayliss first described that an isolated blood vessel responds to an increase in intravascular pressure with a paradoxical decrease in diameter, and vice versa (Bayliss, 1902). Thus was borne the concept of autoregulation of blood flow and, indeed, even today, the assessment of myotonic tone through pressure myography remains the gold standard for measuring blood vessel reactivity (Wilson et al., 2022). A few years later, working on canine heart–lung preparations, Starling and colleagues described the ability of the ventricle to augment its force of contraction as a function of myocardial stretch, allowing it to match cardiac output to increased venous blood return to the heart, the so called Frank–Starling law (Patterson et al., 1914). These are two of the earliest examples of intrinsic mechanisms within the circulatory system to finetune regional function in the face of local physiological perturbations.

Additionally, Bayliss and Starling, through experiments investigating the mechanisms of peristalsis, discovered that a substance released from the duodenum, named secretin, could stimulate pancreatic acid secretion independent of vagal tone (Bayliss & Starling, 1902). Subsequently, in a lecture to the Royal College of Physicians, Starling coined the word 'hormone' to describe the action of secretin and adrenaline (the only other identified hormone at the time) '*These chemical messengers, however, or hormones (in Greek meaning to excite or arouse) as we might call them*' (Starling, 1905). Thus, it became clear that extrinsic factors such as circulating hormones also play important roles in regulating cardiovascular function.

The Physiological Society created the Bayliss Starling Prize Lecture as a joint memorial in 1960, and in 1979 the Bayliss and Starling Society was established. Its main objective was to '*advance education and science by the promotion, for the benefit of the public, the study of the chemistry, physiology, and disorders of central and peripheral regulating peptides and by the dissemination of the results of such study and research*'. It merged with the Physiological Society in 2014 enabling an annual Bayliss Starling Prize Lecture to be awarded. This was awarded in 2023 to Professor Neil Herring whose research group focuses on the role of neuropeptides and neuromodulators in cardiac autonomic regulation.

The autonomic nervous system is now recognised as an important extrinsic regulator of the circulatory system, able to co-ordinate the function of the heart and vasculature. Modern techniques such as tissue clearing allow 3-D imaging of cardiac innervation and reveal a dense network of nerve fibres throughout the myocardium (Hanna et al., 2021; Rajendran et al., 2019). The autonomic nervous system consists of two efferent arms: the sympathetic and parasympathetic systems. In the simplest terms, these two arms exert reciprocal control on cardiac indices acting like an 'accelerator and brake'. Sympathetic tone increases inotropy, chronotropy and dromotropy, whereas parasympathetic tone results in a decrease in these parameters (Kollai & Koizumi, 1979; Levy & Martin, 1989). However, in reality, these are

only part of a complex neurocardiac hierarchy containing efferent motor neurons and afferent sensory neurons, as well as local circuit neurons and interneurons (Armour, 2004). For conceptual purposes, this hierarchy can be considered to consist of three levels: (1) higher cortical centres in the brain stem and the spinal cord; (2) intrathoracic ganglia that receive input from higher cortical centres, afferent sensory neurons and local circuit neurons; and (3) the intrinsic cardiac nervous system (or 'little brain' of the heart) consisting of intrinsic cardiac neurons organised into ganglionic plexi that function as the final beat-to-beat co-ordinator of regional cardiac indices, under the influence of the higher levels of the neurocardiac hierarchy (Armour, 2008; Armour et al., 1997). All levels of this hierarchy are independently able to influence cardiac performance, and collectively work together through multiple interacting feedback loops to provide cardiovascular stability (Ardell & Armour, 2016; Shivkumar et al., 2016). Examples include the arterial baroreceptor reflex, which is able to co-ordinate cardiac output with vasomotor tone and renal water retention to optimise blood pressure control (Kaufmann et al., 2020), and respiratory sinus arrhythmia, which optimises heart lung interactions (Elstad et al., 2015; Giardino et al., 2003).

Historically, neurotransmission was considered to follow Dale's principle, which states that neurons use only a single neurotransmitter (Dale, 1935). In line with this, it was long believed that the parasympathetic system mediates it effects on the heart though ACh activation of cardiac muscarinic G protein coupled receptors (Saternos et al., 2018), whereas sympathetic effects are mediated though noradrenaline acting on alpha- and beta-adrenergic G protein coupled receptors (Lymperopoulos et al., 2013). However, the discovery of non-adrenergic, non-cholinergic nerves in the 1970s has led to a the realisation that neurons can release more than one kind of neurotransmitter, called co-transmission (Burnstock, 1980). A large number of co-transmitters have since been implicated in modulating autonomic signalling such as ATP, nitric oxide (NO), and neuro-peptides such as neuropeptide-Y (NPY), galanin and vasoactive interstitial peptide (Burnstock, 2013).

Our group focusses on such neuropeptides and neuro-modulators that influence cardiovascular autonomic control as potential biomarkers in disease and novel targets for therapeutic intervention. Here, we outline several neuromodulators that may play a key role in cardiovascular pathophysiology. We then focus on the group's current experimental approaches with respect to investigating NPY dynamics and its receptors' signalling pathways in cardiomyocytes, the coronary vasculature and autonomic neurones. Finally, we highlight our recent translational clinical studies which demonstrate the importance of this signalling in acute myocardial infarction (MI) and chronic heart failure (CHF).

## Sympatho-vagal balance

In health, the neurocardiac axis exists in a state of intricate balance maintained through several mechanisms, including intrinsic neuromodulation, paracrine and endocrine effects, as well as via neuropeptides acting as co-transmitters (Goldberger et al., 2019). Some of the key neuromodulators discussed here are outlined in Fig. 1.

**NO and intrinsic neuromodulation.** NO is an important modulator of autonomic neurotransmission. Its synthesis in autonomic neurons is regulated by neuronal NO synthase (nNOS) which has been localised to both cholinergic neurons in the intrinsic cardiac nervous system and efferent sympathetic neurons in the stellate ganglia (Choate et al., 2001; Herring et al., 2002; Paton et al., 2002).

Activation of nNOS in pre-junctional parasympathetic neurons potentates ACh release (Herring et al., 2000). This is mediated through a cGMP-PDE3-dependent pathway, leading to cAMP-PKA-dependent phosphorylation of N-type calcium channels, thus increasing calcium trans-ients and ACh release (Herring & Paterson, 2001). Conversely in sympathetic neurons, nNOS activation reduces noradrenaline release (Choate & Paterson, 1999). Here, NO acts via cGMP-PDE2-dependent reduction of cAMP-PKA activity to reduce neuronal calcium transients and noradrenaline release (Li et al., 2007; Wang et al., 2006, 2007).

**Paracrine and endocrine modulation.** Natriuretic peptides are a group of peptide hormones that have an important role in regulating cardiovascular homeo-stasis. Atrial natriuretic peptide and brain-natriuretic peptide (BNP) are released by cardiomyocytes in the atria and ventricles in response to myocardial stretch, especially in conditions of fluid overload, resulting in elevated plasma levels in conditions such as heart failure. BNP is well established in the diagnosis, assessment of severity and prediction of mortality in CHF (McDonagh et al., 2021). C-type natriuretic peptide (CNP) is primarily expressed in endothelial cells. Whilst circulating plasma concentrations of CNP are also increased in CHF, they are comparatively low compared to atrial natriuretic peptide and BNP (Kalra et al., 2003; Stingo et al., 1992). However, CNP concentrations within cardiac tissue are significantly elevated in CHF, suggesting that it acts locally (Kuwahara, 2021). Natriuretic peptides are considered cardioprotective because of their endocrine effects on vascular tone and renal function, reducing blood pressure and plasma volume (Chinkers et al., 1989; Espiner, 1994). They have also been shown to attenuate cardiac negative remodelling through anti-fibrotic and anti-hypertrophic effects (Lee & Burnett, 2007; Soeki et al., 2005).

Natriuretic peptides also have paracrine effects on autonomic nerves. *In vivo* natriuretic peptides augment vagal tone (Zeuzem et al., 1990). We have shown that BNP acts pre-junctionally to increase ACh release. Similar to nNOS signalling, this is also mediated through cGMP-PDE3-dependent phosphorylation of N-type calcium channels (Herring et al., 2001).

Natriuretic peptides have the opposite effect on sympathetic neurons where they reduce calcium currents and inhibit noradrenaline release. Here, BNP acts on pre-junctional natriuretic peptide receptor (NPR)-A receptors coupled to a cGMP/PKG-dependent pathway (Li et al., 2015), whereas CNP acts through pre-junctional NPR-B receptors (Buttgereit et al., 2016).

Finally, the renin–angiotensin–aldosterone system (RAAS) has important endocrine effects on autonomic nerves (Zucker et al., 2004). Angiotensin II acts centrally on AT1 receptors in the brainstem to augment central sympathetic outflow (Zhu et al., 2002, 2004). Peripherally, AT1 receptor stimulation augments noradrenaline release from sympathetic neurons (Sasaoka et al., 2008). We have shown that AT1 receptors colocalise to sympathetic neurons in the stellate ganglia, and that angiotensin-II acts pre-juntionally to increase noradrenaline release (Herring et al., 2011).

### Neuropeptides and cross-talk

*NPY.* NPY is a 36 amino acid peptide distributed in many parts of the body. It is highly expressed in sympathetic neurons (Lundberg et al., 1983) and is the most abundant neuropeptide in the heart (Gu et al., 1983). Biologically active $NPY_{1-36}$ is generated from pre-pro-NPY through a series of truncations. $NPY_{1-36}$ can be further truncated by dipeptidyl peptidase IV (DPP-4) to give $NPY_{3-36}$, which remains biologically active. In peripheral sympathetic nerve endings, NPY is stored in synaptic vesicles (Ekblad et al., 1984; Fried et al., 1985) and co-released with noradrenaline, particularly during strong or persistent sympathetic activation (Lundberg et al., 1989, 1990).

NPY mediates its action through six known G-protein coupled receptors, of which only Y1, Y2 and Y5 are associated with cardiovascular effects. All three receptors are coupled to inhibitory $G_{\alpha I}$ pathways, resulting in adenylyl cyclase inhibition and reduction in cellular cAMP. Additionally, Y1 is also coupled to a phospholipase C coupled pathway, causing calcium release from intracellular stores via inositol triphosphate (Tan et al., 2018). Generally, Y1 receptors are responsible for post-junctional effects, whereas Y2 receptors mediate inhibitory

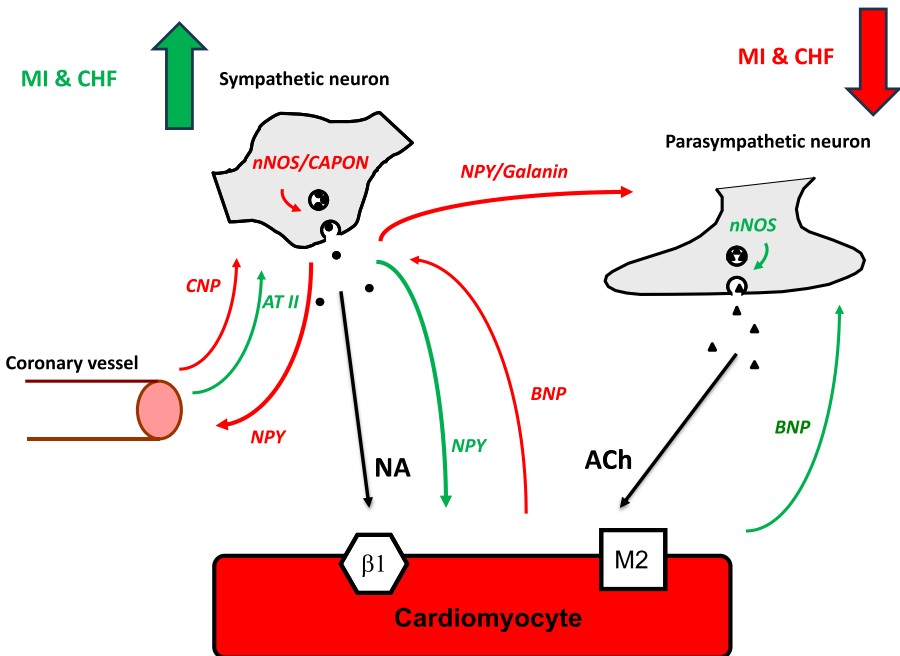

**Figure 1. Schematic demonstrating the various mechanisms involved in regulating autonomic balance at the end organ level**
Black arrows indicate the action of primary neurotransmitters noradrenaline and acetylcholine. Red (inhibition) and green (potentiation) arrows indicate the action of various co-transmitters and hormones in modifying autonomic tone. In diseased states such as chronic heart failure and myocardial infection these mechanisms are perturbed, favouring sympathoexcitation and vagal withdrawal. ACh, acetylcholine; AT II, angiotensin II; $\beta$1, beta 1 adrenergic receptor; BNP, brain-natriuretic peptide; CAPON, carboxy-terminal PDZ ligand of nNOS; CHF, chronic heart failure; CNP, C-type natriuretic peptide; M2, type 2 muscarinic receptor; MI, myocardial infarction; NA, noradrenaline; nNOS, neuronal nitric oxide synthase; NPY, neuropeptide Y.

pre-junctional neuromodulator effects (Westfall, 2004). The role of Y5 is less well understood, but it appears to function as a modulator of Y1 and Y2 activity. The N-terminal portion of $NPY_{1-36}$ determines its binding affinity to the Y1 receptor. Conversation of $NPY_{1-36}$ by DPP-4 to $NPY_{3-36}$ results in a loss of this N-terminal potion, leading to a loss of Y1 affinity but retained high affinity for Y2 and Y5 receptors. DPP-4 can thus be considered as a molecular switch that shifts NPY activity from Y1 to Y2 mediated effects (Zukowska et al., 2003).

Potter (1985, 1987) demonstrated that prolonged sympathetic activation reduces parasympathetic activity. This cross-talk effect remains even in the presence of adrenergic blockade. Furthermore, it can be replicated through exogenous NPY application even in the absence of sympathetic stimulation.

We have shown that the Y2 receptor co-localises with choline acetyltransferase containing neurons in the sinus node. Furthermore, NPY acts on these Y2 receptors to reduce ACh release from parasympathetic nerve terminals, through a protein kinase C-dependent pathway (Herring et al., 2008). The importance of this pathway in mediating autonomic cross-talk is demonstrated by the fact that genetic knockout of the Y2 receptor in mice, or selective antagonism of the Y2 receptor, reduces vagal bradycardia post synaptic stimulation (Ilebekk et al., 2005; Smith-White et al., 2002).

NPY also acts as a negative-feedback modulator of sympathetic transmission. Pre-junctional Y2 receptors in sympathetic neurons reduce noradrenaline release through a protein kinase C-dependent pathway (Martin et al., 1998; Wahlestedt et al., 1986; Westfall et al., 1987). Conversely, circulating noradrenaline increases NPY release via neuronal beta receptors, which can be blocked by propranolol (van Weperen et al., 2025).

*Galanin.* Galanin is another neuropeptide which is distributed throughout the central and peripheral nervous system (Crawley, 1995). Centrally galanin has modulatory effects on sympathetic outflow from the nucleus of the solitary tract (Díaz-Cabiale et al., 2010). It is also expressed peripherally in sympathetic neurons (Kummer, 1987), particularly in response to neuronal injury (Schreiber et al., 1994; Strömberg et al., 1987). In humans, galanin concentrations increase with sympathetic activation during exercise (Legakis et al., 2000). Infusion of galanin increases resting heart rate and supresses respiratory sinus arrhythmia, suggesting anti-vagal effects (Carey et al., 1993). We have shown that galanin co-localises with sympathetic neurons in the stellate ganglion, and that the GalR1 receptor co-localises to parasympathetic nerve terminals in the sinus node. Indeed, galanin is released following high frequency sympathetic stimulation, and attenuates ACh release via a PKC-dependent pathway, independent of NPY (Herring et al., 2012).

## Autonomic imbalance in heart disease

The intricate balance within the autonomic nervous system is lost in cardiac disease (Corr & Gillis, 1978). The hallmark of a wide range of cardiac diseases, irrespective of aetiology, is sympathetic activation and parasympathetic withdrawal (Florea & Cohn, 2014; Malliani et al., 1969).

In animal models of CHF, there is increased gain in sympathetic reflexes leading to enhanced sympathetic tone (Ishise et al., 1998; Ma et al., 1997; Motte et al., 2005; Sun et al., 1999). Direct neuronal recordings in dogs and sheep show persistently elevated sympathetic activity in the stellate ganglia following MI (Han et al., 2012; Jardine et al., 2005) and increased renal sympathetic nerve activity in CHF (DiBona et al., 1995; Ramchandra & Barrett, 2015; Zucker & Wang, 1991). Single-cell patch clamp recordings of sympathetic neurons in the stellate ganglia of spontaneously hypertensive rats reveal a hyper-active phenotype, manifest as an increased firing rate and a depolarised resting membrane potential (Davis et al., 2020). These changes even precede the onset of hypertension, suggesting a causal role in pathogenesis (Li et al., 2012; Shanks et al., 2013).

Similar findings are seen in humans. Acute MI and CHF are both associated with increased plasma and urinary catecholamines (Corr & Gillis, 1978; Gazes et al., 1959; Nuzum & Bischoff, 1953) and noradrenaline spillover from the heart (Meredith et al., 1993), which are strongly predictive of adverse outcomes (Cohn et al., 1984). Patients with CHF, hypertension and MI demonstrate increased muscle sympathetic nerve activity according to microneurography studies (Grassi, 2009; Grassi et al., 1995; Gronda et al., 2014; Huang et al., 2022; Leimbach et al., 1986). In patients with CHF, or following acute MI, there is also a well-documented alteration in the arterial baroreflex (Eckberg, 1997; Ferguson et al., 1992; Olivari et al., 1983; Thames et al., 1993) characterised by disinhibition of sympathetic output, and reduced vagal tone (Eckberg et al., 1971; Saul et al., 1988), which is predictive of mortality in these patients (De Ferrari et al., 2007; La Rovere et al., 1998, 2009; Pinna et al., 2005)

In addition to sympathetic activation, heart disease is associated with extensive neurohormonal activation, including elevated plasma levels of natriuretic peptides, catecholamines and the RAAS (Cohn et al., 1984; Sohaib et al., 2013). In particular, increases in angiotensin II further entrenches autonomic imbalance by maintaining central sympathetic activation via AT1 receptors in the rostral ventrolateral medulla (Zucker et al., 2004).

## Remodelling of the autonomic nervous system in cardiac disease

The observed autonomic imbalance in cardiac disease is accompanied by functional and morphological

remodelling within the neurocardiac hierarchy. This is outlined in detail in the translational white paper published in this issue of *The Journal of Physiology*. Here, we focus specifically on changes in neuromodulator expression and remodelling.

Cardiac disease is associated with changes in the nNOS pathway that predispose to sympathetic activation. In guinea-pigs exposed to chronic intermittent cardiac ischaemia, enhanced sympathetic activity is associated with reduced nNOS expression (Mohan et al., 2001). Similarly, the heightened sympathetic activity seen in spontaneously hypertensive rats (Shanks et al., 2013) is associated with reduced expression of nNOS in the stellate ganglia, and can be reversed by targeted transgene nNOS delivery (Li et al., 2007). In Langendorff perfused rat hearts, inhibition of nNOS attenuates the anti-fibrillatory action of the parasympathomimetic carbamylcholine, demonstrating the clinical importance of this pathway (Kalla et al., 2016).

Furthermore, we recently showed that reductions in CAPON (carboxy-terminal PDZ ligand of nNOS) expression, also known as nNOS activator protein (NOS1-AP), led to sympathetic activation (Lu et al., 2015). CAPON interacts with nNOS to escort it to specific protein targets within the cell (Jaffrey et al., 2002). This is important because the action of NO within a cell is determined by the location of nNOS as a result of the short half-life and high reactivity of NO as a free radical gas. In spontaneously prehypertensive rats, CAPON co-localises to sympathetic neurons in the stellate ganglia, although its expression is reduced compared to wild-type controls. Delivery of CAPON to sympathetic neurons using a specific adenovirus vector can correct the heightened sympathetic phenotype observed in these animals. It is worth noting that genome-wide association studies have highlighted single nucleotide polymorphisms in *NOS1AP* associated with QT interval variation (Arking et al., 2006) and sudden cardiac death (Eijgelsheim et al., 2009). Polymorphisms in *NOS1AP* have also been identified as risk modifiers for arrhythmic events and sudden cardiac death in patients with LQTS type 1 (Crotti et al., 2009) where sympathetic drive is a key trigger (Schwartz et al., 2001).

Sympathetic neurons also show an altered ion channel expression profile which predisposes to increased excitability. Single cell mRNA sequencing in spontaneously hypertensive rats reveals reduced expression of subunit genes associated with the M current (Davis et al., 2020), an inhibitory potassium current that has important influence on neuron resting potential and restricting neuron firing (Wladyka & Kunze, 2006). Similarly, in rats with ischaemic cardiomyopathy, there is enhanced sympathetic excitability due to increased N-type calcium currents (Tu et al., 2014).

Finally, PDE2A activity is upregulated in patients with heart failure (Mehel et al., 2013; Mongillo et al.,

2006) and in the stellate ganglia of spontaneously hypertensive rats (Li et al., 2015) or patients with refractory arrhythmia (Liu et al., 2018). PDE2A is a dual substrate esterase able to breakdown both cAMP and cGMP, with its selectivity depending on cAMP and cGMP coactivation (Zaccolo & Movsesian, 2007). In cardiomyocytes, activation of PDE2A leads to reduction in cAMP and thus a blunted response to beta-adrenergic stimulation (Mehel et al., 2013; Mongillo et al., 2006). In sympathetic neurons, however, PDE2A appears to favour hydrolysis of cGMP while cAMP levels are less affected. In spontaneously hypertensive rats, increased PDE2A expression in sympathetic neurons depletes cGMP levels and reduces PKG activity, leading to increased calcium currents and augmented noradrenaline release. This upregulation of PDE2A also leads to a blunted response to BNP, thus limiting its sympathoinhibitory effects (Li et al., 2015; Liu et al., 2018).

The importance of natriuretic peptide mediated sympathoinhibition is illustrated in a rat model of sympathetic neuron-specific overexpression of a negative mutant of NPR-B (Buttgereit et al., 2016). In these animals, the blunted sympathoinhibitory action of CNP results in heightened sympathetic tone, leading to raised blood pressure, resting tachycardia and impaired left ventricular systolic function. Importantly, the fact that increased sympathetic tone results in heart failure even in the absence of cardiac injury demonstrates a causal role for sympathoexcitation in the pathophysiology of heart failure. Similar observations have been made in mice where genetic disruption alpha2 adrenergic receptors results in sympathoexcitation and heart failure before 4 months of age (Brum et al., 2002; Hein et al., 1999).

Cardiac disease is also associated with morphological remodelling within the neurocardiac hierarchy. In both ischaemic and non-ischaemic heart failure, nuclear imaging consistently demonstrates reduced uptake of radiolabelled catecholaminergic tracers in the heart, consistent with denervation (Parthenakis et al., 2002). MI causes sympathetic denervation, which can even extend beyond the infarction zone, particularly in areas distal to the infarcted territory (Barber et al., 1983; Dae et al., 1991, 1995). Over time, there is some neuronal regeneration characterised by sympathetic nerve sprouting at the infarct border zone (Cao et al., 2000; Li & Li, 2015). The resultant hyperinnervation adjacent to denervated infarcted myocardium causes regional heterogeneity in sympathetic innervation, which is highly pro-arrhythmic (Cao et al., 2000; Dajani et al., 2023; Vaseghi et al., 2012), whereas homogeneity of stimulation is anti-arrhythmic (Tomek et al., 2019). Furthermore, denervation leads to supersensitivty to catecholamines (Tapa et al., 2020). MI also leads to morphological changes within the intrinsic cardiac nervous system, such as neuronal enlargement, which are associated with reduced functional network

connectivity and altered responses to cardiac pre-load and pacing (Rajendran et al., 2016).

Finally, there is well documented neural remodelling within the stellate ganglia. In animal models of CHF, there is an increase in synaptic density and size of sympathetic neurons (Han et al., 2012), as well as increased numbers of adrenergic and NPY containing neurons (Ajijola et al., 2015). Inflammation appears to play an important role in stellate ganglion remodelling. In rats, myocardial infarction is associated with macrophage expansion and expression of pro-inflammatory cytokines tumour necrosis factor-alpha and interleukin-1B in the stellate ganglia (Zhang et al., 2021) and stellate ganglion injection of interleukin-1B leads to increased sympathetic activity and reduced nNOS expression (Wang et al., 2017).

Stellate ganglion remodelling is also well documented in humans. In stellate ganglia of patients with cardio-myopathy or recurrent ventricular arrhythmias, sympathetic neurons demonstrate signs of oxidative stress and adrenergic profiles in keeping with sympathetic activation. There are also pro-inflammatory changes such as neutrophil and T-cell infiltration, as well as activation of satellite microglia (Ajijola et al., 2017; Rizzo et al., 2014).

## Causal role of autonomic imbalance in cardiac disease

Sympathetic activation is initially a physiological response to maintain cardiac output and blood pressure following acute cardiac injury (Zucker et al., 2004, 2012). However, in the long-term, persistent sympathetic hyperactivation becomes maladaptive, contributing to the progression of cardiac disease and predisposing to ventricular arrhythmias and sudden cardiac death (Herring et al., 2019a; Lymperopoulos et al., 2013; Vaseghi & Shivkumar, 2008).

In rodents, increased sympathetic activation leads to heart failure and fibrosis even in the absence of direct cardiac injury (Brouri et al., 2004; Brum et al., 2002; Buttgereit et al., 2016; Hein et al., 1999). Conversely, attenuating sympathetic activation is protective against negative remodelling (Wang et al., 2014; Yoshie et al., 2020) and reduces myocardial inflammation (Ziegler et al., 2017) following myocardial infarction. Chronic adrenergic stimulation of cardiomyocytes leads to desensitisation and downregulation of beta-adrenergic receptors via G-protein-coupled receptor kinase 2, resulting in progressive loss of inotropic reserve (Eschenhagen, 2008; Huang et al., 2011). Furthermore, chronic sympathetic stimulation increases energy consumption (Spindler et al., 2003) and alters ventricular excitation–contraction coupling through alteration of cellular calcium handing (Joca et al., 2020). Ultimately, chronic sympathetic overstimulation causes calcium over-load and the formation of reactive oxygen species, leading to cardiomyocyte hypertrophy and apoptosis (Communal et al., 1998; Fu et al., 2004).

Sympathetic stimulation is also highly pro-arrhythmogenic. In animal models, stimulation of the stellate ganglia decreases fibrillatory threshold and induces ventricular arrhythmia (Harris et al., 1971; Schwartz & Vanoli, 1981; Schwartz et al., 1985). In patients with MI, arrhythmic risk increases with sympathetic activation (Huang et al., 2022). Beta-adrenergic stimulation increases intracellular calcium loading in myocytes, predisposing to delayed afterdepolarisations, which are triggers for most pathological arrhythmias (Lubbe et al., 1992; Shiferaw et al., 2012; Tsien et al., 1986). Indeed, increasing sympathetic activation to the heart through stellate ganglion stimulation has been shown to induce delayed afterdepolarisations *in vivo* (Priori et al., 1988). The substrate required to sustain arrhythmias is usually in the form of re-entry around an area of anatomical or functional block. In general, slow conduction velocities and shorter action potential durations are pro-arrhythmic because they allow faster myocardial recovery from previous depolarisation, thus enabling reactivation as part of a re-entry circuit. Beta-adrenergic stimulation in known to increase $I_{KS}$ (slow outward potassium current), which results in shorter action potential duration (Sanguinetti et al., 1991). Furthermore, because of regional differences in sympathetic innervation, sympathetic activation has been shown to produce heterogeneity in cardiac myocyte electrical properties in the heart which predisposes to arrhythmia (Ng et al., 2009). *In vivo* stellate ganglion stimulation increases the time interval from the peak to the end of the electrocardiographic T wave, which is considered as a marker of dispersion of repolarisation in the heart, and is an independent predictor for risk of sudden cardiac death (SCD) (Yagishita et al., 2015). This effect is further accentuated by sympathetic nerve sprouting and supersensitivity following myocardial infarction (Cao et al., 2000).

Sympathetic activation also reduces parasympathetic tone through autonomic cross-talk. Vagal nerve stimulation is generally cardioprotective, increasing the fibrillation threshold, preventing negative remodelling and having anti-inflammatory effects (Beaumont et al., 2015; Del Rio et al., 2008; Díaz et al., 2020; Nash et al., 2001; Shinlapawittayatorn et al., 2013; Vanoli et al., 1991). By mitigating these protective effects, sympathetic activation further contributes to disease progression.

Overall, it is intuitive that pharmacological sympathetic blockade is cardioprotective. Indeed, beta-blockers reduce adverse remodelling, are anti-arrhythmic, and ultimately reduce mortality post MI and during CHF, as shown in randomised controlled trials such as CIBIS-II (1999) and MERIT-HF (1999). Similarly, pharmacological inhibition

of the RAAS reduces sympathetic tone (Cody et al., 1982), increases vagal tone (Osterziel et al., 1990; Townend et al., 1992) and leads to improved survival in heart failure and following myocardial infarction (Pfeffer, 1995; Study, 1993; Yusuf et al., 1991). However, despite optimal medical therapy, the risk of arrhythmia and death in heart failure remains high, at ∼7–10% annually (Bardy et al., 2005; Moss et al., 2002). Increasingly, it is becoming clear that these residual adverse effects of sympathoexcitation are mediated by co-transmitters such as NPY. Indeed, sympathoexcitation through stellate ganglion stimulation results in pro-arrhythmic electrophysiological effects, even in the presence of supra-clinical doses of beta-blockers. However, this effect is blocked with the addition of a selective Y1 antagonist (Hoang et al., 2020).

## NPY concentrations are elevated in heart disease

It has recently been shown that the heart is innervated predominantly by NPY-expressing sympathetic neurons. Single cell mRNA sequencing of mouse stellate ganglia has identified a subpopulation of sympathetic neurons that express NPY and poses a distinct transcriptomic profile associated with increased excitability, a finding that has also been validated in humans. Viral retrograde tracing shows that that the heart is predominantly innervated by these NPY expressing neurons. Furthermore, selective targeting of these neurons in NPY-hrGFP mice confirms they possess distinct electrophysiological properties, and that they are required to achieve maximal cardiac sympatho-excitation (Sharma et al., 2025).

Indeed, it has long been observed that plasma NPY-like activity is elevated in states of sympathetic hyper-activity such as CHF and MI (Maisel et al., 1989; Omland et al., 1994). Stellate ganglia from patients with heart failure show decreased NPY immunoreactivity despite unchanged NPY mRNA expression, implying increased release of NPY to the heart (Ajijola et al., 2020). Modern assays enable measurement of NPY levels without cross-reactivity to similar peptides. As part of the Oxford Acute Myocardial Infarction (OxAMI) study, we confirmed that peripheral venous levels of NPY are significantly elevated in patients presenting with ST-elevation myocardial infarction (STEMI) and remained elevated for at least 48 h despite successful culprit vessel percutaneous coronary intervention (PCI). Furthermore, NPY elevation was greatest in patients with angiographic no-reflow despite successful PCI, and those with microvascular dysfunction as measured by coronary flow reserve and index of microvascular resistance (Cuculi et al., 2013).

However, because the liver and gut contribute significantly to NPY plasma levels, it remained unclear to what extent peripheral venous levels of NPY reflect cardiac NPY release (Morris et al., 1997). By measuring coronary sinus (CS) NPY levels, we were able to confirm that NPY elevations seen in STEMI where indeed a result of cardiac release (Herring et al., 2019b). We showed that, in addition to worse microvascular function, patients with higher cardiac levels of NPY experience more ventricular arrhythmias within 48 h of index PCI, have larger infarct size and end up with worse left ventricular function at 6 months post STEMI (Herring et al., 2019b; Kalla et al., 2020).

Importantly, NPY is also a predictor of mortality in both MI and CHF (Fig. 2). Following STEMI, plasma NPY levels are independent predictors of heart failure and death (Gibbs et al., 2022). In heart failure patients undergoing cardiac resynchronisation implantation, elevated CS NPY concentrations are associated with major adverse cardiac events (Ajijola et al., 2020). Similarly, in a cohort of 833 CHF patients, peripheral NPY concentrations are associated with all cause mortality and cardiovascular death, independent of serum BNP levels (McDowell et al., 2024). Interestingly, in heart failure, the degree of NPY elevation is not associated with severity of left ventricular systolic impairment or heart failure hospitalisations. Rather, the observation that NPY selectively predicts mortality in these patients suggests it plays a role in arrhythmogenesis, which is the most common mode of death in patients with structural heart disease (Mitrani & Myerburg, 2016).

Another important observation is that elevations in CS NPY concentrations are generally mirrored by elevations in peripheral venous concentrations, supporting the use of peripheral venous NPY levels as a surrogate for cardiac NPY levels. This is clearly illustrated by comparing CS and peripheral venous NPY concentrations in patients across a spectrum of coronary artery disease and heart failure, as seen in Fig. 3.

Finally, it is worth noting that NPY has also been implicated in the pathogenesis of takotsubo syndrome where peripheral venous levels are acutely elevated (Szardien et al., 2011; Wittstein et al., 2005). Although the pathophysiology of takotsubo is still poorly understood, it is generally accepted that transient left ventricular impairment results from a surge of sympathetic stimulation of the heart (Ghadri et al., 2018; Lyon et al., 2021; Pelliccia et al., 2017). The resultant mass release of catecholamines and NPY then has direct cardio-toxic effects, and through vasoconstrictive effects on the microvasculature may lead to ischaemic stunning of the myocardium (Galiuto et al., 2010). In a mouse model of takotsubo syndrome, there is increased NPY expression in the stellate ganglia. Furthermore, elimination of NPY expression using small interfering RNA reduced the incidence of takotsubo in these mice, suggesting a causal role of NPY in this mouse model (Arai et al., 2022).

## Pathophysiological role of NPY in heart disease

NPY can directly influence cardiac and vascular function through its Y receptors, independent of adrenergic activity.

NPY has positive inotropic effects on cardiomyocytes via its Y1 receptor. This is mediated through a phospholipase C-dependent pathway leading to increased calcium transients (del Puy Heredia et al., 2005). NPY is also a potent vasoconstrictor. Infusion of NPY into human coronary arteries is able to induce ischaemic chest pain and ST elevation, even in the absence of epicardial coronary constriction (Clarke et al., 1987). Immuno-histology confirms extensive expression of Y1 receptors on coronary microvascular smooth muscle cells. We have demonstrated that NPY acts directly on micro-vascular smooth muscle cells via its Y1 receptor to cause dose-dependent vasoconstriction. This vasoconstrictive effect causes an increase in coronary vascular resistance in Langendorff perfused rat hearts. In the context of myocardial ischaemia, this leads to increased infarct size presumably though microvascular vasoconstriction in the vulnerable peri-infarct zone (Herring et al., 2019b).

We have also demonstrated that NPY is pro-arrhythmic, independent of beta-adrenergic stimulation (Kalla et al., 2020). In isolated Langendorff heart preparations, stellate ganglion stimulation causes significant NPY release to the heart. Even in the presence of combined beta- and alpha-adrenergic blockade, stellate ganglion stimulation causes an increase in ventricular calcium transients and reduced ventricular fibrillation threshold. This effect is abolished by selective Y1 blockade. Similar effects are seen in a mouse models of subarachnoid haemorrhage, a condition associated with increased risk of ventricular arrhythmias (Michael Frangiskakis et al., 2009), where upregulation of Y1 receptors is associated with a reduction in ventricular fibrillation threshold (Chen et al., 2023).

NPY also has trophic effects on cardiomyocytes. *In vitro* administration of NPY to isolated ventricular cardiomyocytes increases protein synthesis and causes hypertrophy through $Ca^{2+}$/calmodulin-dependent calcineurin signal pathways (Chen et al., 2005; Millar et al., 1994; Nicholl et al., 2002). This is also true *in vivo* where chronic administration of NPY causes cardiac dysfunction and hypertrophy (Zhang et al., 2015). NPY has also been shown to disrupt energy metabolism and mitochondrial integrity, adversely affecting cardio-myocyte viability (Hu et al., 2017; Luo et al., 2015). Transgenic mice with overexpression of NPY are more prone to doxorubicin mediated cardiomyopathy, whereas NPY knockout rats have improved cardiac function and reduced apoptosis following MI (Huang et al., 2019). However, unexpectedly, a different study using NPY knockout mice demonstrated larger infarct size and worsening cardiac function following MI in NPY knockout mice (Qin et al., 2022), leading to the suggestion that NPY might be cardioprotective in their mouse model. The cause of this discrepancy is unclear but might be a result of different co-transmitter profiles in mice and rats. Interestingly in wild-type mice, high concentrations of NPY remained detrimental.

**Figure 2. Kaplan–Mayer plots**
Kaplan–Mayer plots demonstrating cumulative incidence of outcomes in STEMI ($n = 163$) and heart failure ($n = 833$) patients according to baseline plasma NPY concentrations. Reproduced from Gibbs et al. (2022) and McDowell et al. (2024). NPY, neuropeptide-Y; STEMI, ST-elevation myocardial infarction.

It appears that the effect of NPY on cardiac remodelling depends on the local context, such as the ratio of Y1 to Y2 receptors and DPP-4 activity. *In vitro*, selective stimulation of Y1 and Y2 receptors has opposite effects on inotropy. Y1 simulation leads to positive inotropic responses, whereas Y2 activation opposes the inotropic effects of beta-adrenoreceptor stimulation (Allen et al., 2006). The Y2 receptor pathway may be cardioprotective. Selective activation of Y2 by $NPY_{3-36}$ promotes angiogenesis (Lee et al., 2003; Saraf et al., 2016) and has been demonstrated to improve cell survival and reduce fibrosis in a swine model of myocardial ischaemia (Matyal et al., 2013; Robich et al., 2010). Conversely, Y1 signalling reduces cardiomyocyte viability and induces apoptosis (Huang et al., 2019). In patients with arrhythmogenic cardiomyopathy, there is increased Y1 receptor expression in cardiac mesenchymal stromal cells, leading to increased adipogenesis and fibrofatty myocardial replacement (Stadiotti et al., 2021). In CHF, there is a shift from Y1 to Y2 expression, possibly as a compensatory response to chronic sympathetic activation (Callanan et al., 2007).

DPP-4 plays an important regulatory role on the action of NPY. For example, during exercise recovery, DPP-4 rapidly converts $NPY_{1-36}$ to $NPY_{3-36}$ as the sympathetic tone declines (Eugster et al., 2022). DPP-4 is expressed in human left ventricular cardiomyocytes and its expression is reduced in heart failure (Vörös et al., 2022). The clinical importance of DPP-4 activity has been highlighted by the cardiotoxic effects seen in diabetic patients treated with DPP-4 inhibitors (Scirica et al., 2013). Although inhibition of DPP-4 prevents the degradation of incretins facilitating insulin secretion (Thornberry & Gallwitz, 2009), it also prevent conversion of $NPY_{1-36}$ to $NPY_{3-36}$, thus favouring cardiotoxic Y1 receptor signalling.

Finally, NPY has well documented immunomodulatory effects (Chen et al., 2020). It can influence macrophage differentiation between M1 and M2 phenotypes, which has important implications for cardiac remodelling (Nahrendorf & Swirski, 2013). However, these effects are again dependent on NPY concentration and DPP-4 activity (Chen et al., 2020; Dimitrijević et al., 2008).

### Therapeutic and diagnostic implications

Autonomic modulation is an emerging strategy for the treatment of heart failure and ventricular arrhythmia (VA). Reducing sympathetic tone through deep sedation has been shown to be effective with respect to reducing VAs in patients refractory to anti-arrhythmic medication

**Neuropeptide Y Blood Concentrations by Clinical Presentation**
Peripheral Venous vs Coronary Sinus

**Figure 3. Bar plot comparing mean coronary sinus and peripheral venous NPY concentrations across a spectrum of cardiac disease**
'Non-flow limiting CAD' are patients without flow limiting epicardial coronary artery disease at the time of invasive coronary angiography. 'Flow-limiting CAD' are patients who required percutaneous coronary intervention for significant epicardial coronary artery stenosis (including angina and non-ST elevation myocardial infarction). Reproduced from Herring et al. (2019), Ajijola et al. (2020), Gibbs et al. (2022) and McDowell et al. (2024). CAD: coronary artery disease, CHF: chronic heart failure, NPY: neuropeptide Y, STEMI: ST-elevation myocardial infarction.

(Bundgaard et al., 2020). Percutaneous injection of local anaesthetic into the epidural space at level T1–T4 temporarily reduces sympathetic outflow to the heart and can be used to manage refractory arrhythmias in patients with structural heart disease (Bourke et al., 2010; Do et al., 2017). Stellate ganglion blockade through percutaneous local anaesthetic injection has similar effects (Tian et al., 2019).

A more permanent form of cardiac sympathetic denervation (CSD) can be achieved through surgical removal of the stellate ganglia. Surgical CSD is not a novel concept and pre-dates coronary artery bybass grafting by several decades. Its application for treatment of angina was first suggested in 1899 (Francois-Franck, 1899). This inspired Jonnesco in 1916 to perform surgical removal of the left stellate ganglion in a patient with intractable angina, abolishing both anginal attacks and ventricular arrhythmia (Jonnesco, 1921). Subsequently, the surgical technique has been refined and now involves resection of only the lower half of the stellate ganglion (to reduce incidence of Horner's syndrome) and the T2–T4 sympathetic ganglia (Bourke et al., 2010; Vaseghi et al., 2014), conducted as a video-assisted thoracoscopic procedure. Studies have established the use of surgical CSD for prevention of sudden cardiac death in patients with long QT syndrome (Schwartz et al., 1991) and catecholaminergic poly-morphic ventricular tachycardia (Wilde et al., 2008), where it is now guideline-endorsed with a similar level of recommendation as implantable cardioverter defibrillator (ICD) implantation (Zeppenfeld et al., 2022). This concept has been further expanded by Vaseghi et al. (2017), who demonstrated that bilateral surgical CSD is safe and effective at preventing ventricular arrhythmia in patients with structural heart disease. Indeed, it is strikingly effective, with metanalysis demonstrating upward of 60% reduction in VAs in patients in whom arrhythmias are otherwise refractory to all other available treatments including maximally tolerated doses of betablockade (Shah et al., 2019). This observation may be related to the fact that CSD reduces not only noradrenergic drive to the heart, but also NPY release, thus also eliminating the arrhythmic effects of NPY (Hoang et al., 2020; Kalla et al., 2020). There is also limited evidence that bilateral surgical CSD can improve cardiac function and symptoms in patients with systolic heart failure (Conceição-Souza et al., 2012) although these finding are hypothesis generating only, and require larger clinical trials to be investigated further.

Sympathetic tone can also be reduced through renal denervation (Hering et al., 2014). This has been shown to improve blood pressure control in several sham-controlled clinical trials (Azizi et al., 2023, 2024). Furthermore, in animal models, renal denervation increases ventricular fibrillary threshold and reduces VAs

(Ye et al., 2022; Zhang et al., 2018), and such findings have been replicated in case series of patients with refractory ventricular arrhythmia (Armaganijan et al., 2015).

Direct vagal nerve stimulation has shown promise in preventing arrhythmia and adverse cardiac remodelling in animal models (Myers et al., 1974; Nash et al., 2001; Zhang et al., 2009) and, although clinical benefits in heart failure patients were not seen in either the NECTAR-HF or INOVATE-HF trials (Gold et al., 2016; Zannad et al., 2015), the ANTHEM-HF study did demonstrate improvements in symptoms and echocardiographic parameters at both 6 months and 12 months of follow up compared to baseline (Premchand et al., 2014, 2016). The larger randomised controlled ANTHEM-HFrEF Pivotal trial (NCT03425422) is yet to report (Konstam et al., 2019). Vagal tone can also be increased through percutaneous stimulation of the auricular branch of the vagus nerve at the tragus (Jiang et al., 2020). In patients with STEMI, tragal stimulation has been shown to reduce inflammatory markers post infarct and lead to reduced VAs and better myocardial function (Yu et al., 2017). In patients with paroxysmal atrial fibrillation, tragus stimulation reduces inflammatory cytokines and supresses atrial fibrillation and NPY levels (Stavrakis et al., 2015, 2020).

Future therapies selectively targeting signalling pathways within the neurocardiac hierarchy could also be used to re-establish autonomic balance. PDE2A is emerging as an important modulator of autonomic balance through its ability to modulate cAMP and cGMP levels in pre-junctional sympathetic neurons. As already described, increased PDE2A activity favours cGMP breakdown, resulting in increased calcium transients and noradrenaline release in sympathetic neurons, and blunting of the sympathoinhibitoy effect of natriuretic peptides (Li & Paterson, 2016; Li et al., 2015). Indeed, PDE2A activity is elevated in the stellate ganglia of spontaneously hypertensive rats and patients with recurrent ventricular arrhythmias. Pharmacological inhibition of PDE2A and noradrenergic neuron-specific gene transfer of a non-functional form of PDE2A restores cGMP mediated inhibition of noradrenaline release, highlighting this as a potential therapeutic target to reduce sympathetic tone (Liu et al., 2018).

Targeting the nNOS signalling pathway to restore autonomic balance has shown promise in animal models. Targeted gene transfer of nNOS to parasympathetic neurons increases parasympathetic activity in pigs (Heaton et al., 2005) and guinea-pigs (Mohan et al., 2002), and also restores parasympathetic function in spontaneously hypertensive rats (Heaton et al., 2007) and guinea pigs following acute MI (Dawson et al., 2008). Conversely, noradrenergic specific gene transfer of nNOS to sympathetic neurons decreases sympathetic neurotransmission in rats (Wang et al., 2006, 2007)

and rescues impaired NO–cGMP signalling to reduce $Ca^{2+}$-dependent noradrenaline release in a rat model of sympathetic hyperactivity (Li et al., 2007, 2013). Similarly, delivery of CAPON to sympathetic neurons using a specific adenovirus vector has been shown to correct the heightened sympathetic phenotype of spontaneously hypertensive rats (Lu et al., 2015).

Selective modulation of NPY signalling has also shown therapeutic potential in animal models. Stimulation of Y2 receptors may be an emerging technique to encourage angiogenesis following MI. It has recently become possible to deliver $NPY_{3-36}$ to areas of oxidative stress by incorporating it into $H_2O_2$ responsive copolyoxalate containing vanillyl alcohol (i.e. PVAX) nanoparticles. In mouse models of limb and myocardial ischaemia, this resulted in significantly improved blood flow and cardiac function compared to controls (Eshun et al., 2017; Mahmood et al., 2020). Similarly, in pig models of chronic cardiac ischemia, infusion of $NPY_{3-36}$ resulted in improved cardiac blood flow, restoration of fatty acid metabolism, reduced fibrosis and improved left ventricular function (Matyal et al., 2013; Robich et al., 2010). Conversely, selective antagonism of Y1 receptors in pigs has been shown to prevent pro-arrhythmic electro-physiological changes without adverse effects on inotropy (Hoang et al., 2020).

Finally, biomarkers such as NPY could also vastly improve the ability to predict SCD. Current methods are crude and, in CHF, primarily based on left ventricular function, previous ventricular arrhythmias and myo-cardial scar burden (Zeppenfeld et al., 2022). These methods are imperfect and predicting the risk of SCD remains challenging (Deyell et al., 2015). Although ICDs improve overall mortality on a population level (Bardy et al., 2005; Moss et al., 2002), many individuals never go on to require ICD therapy. Such individuals, however, are subjected to morbidity from device related complications and inappropriate device therapies (Ezzat et al., 2015; Hawkins et al., 2018). Elevated peripheral venous NPY levels are strongly associated with mortality following MI and CHF (Fig. 2) (Gibbs et al., 2022; McDowell et al., 2024). Importantly peripheral venous NPY levels appear to be an acceptable surrogate for cardiac NPY release (Fig. 3), making this a practical clinical tool to improve risk stratification. Indeed, the prognostic value of peripheral venous NPY concentrations in STEMI has recently been independently validated (Tiller et al., 2024). Whether additional prognostic information can be gained from investigating NPY dynamics in CHF patients during exercise remains to be determined. Numerous other measures of autonomic activity such as heart rate variability, heart rate recovery after exercise, abnormal baroreflex responses and altered sympathetic innervation on nuclear imaging have also been shown to independently predict mortality and SCD in patients with cardiac disease (Cole et al., 1999; De Ferrari et al., 2007; Fallavollita et al., 2017; La Rovere et al., 1998).

## Conclusions

The autonomic nervous system is an important modulator of cardiac function, and normally exists in a state of balance between its parasympathetic and sympathetic limbs. The hallmark of cardiac disease is a loss of this intricate balance, leading to chronic hyperactivation of sympathetic output, which directly contributes to disease progression. Although beta-blockers improve mortality in these conditions by preventing the action of noradrenaline, a substantial residual risk remains. This can be explained by the existence of co-transmitters such as NPY, which can independently influence cardiac remodelling and arrhythmic risk.

Neuromodulatory interventions to limit sympathetic activation have recently been shown to be effective treatment for some patients with resistant hypertension or recurrent ventricular arrhythmias. This remains an area of active research, with ongoing randomised clinical trials recruiting patients with heart failure and high arrhythmic risk (ClinicalTrials.gov. Identifier NCT01013714; Chin et al., 2017; Zhouting et al., 2023). Interventions to selectively target autonomic co-transmission and neuro-nal signalling pathways have shown promise in animal models but are not yet ready for human use. Finally, biomarkers such as NPY may prove to be a vital tool with respect to improving prediction of SCD and identifying patients most probably benefiting from ICD therapy.

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

## Additional information

### Competing interests

The authors declare that they have no competing interests.

## Author contributions

B.B. drafted the manuscript. All authors were involved in the editing and proof reading of the final manuscript. All authors have approved the final version of the manuscript submitted for publication. All persons who meet authorship criteria are listed as authors, and all authors certify that they have participated sufficiently in the work to take public responsibility for the content in line with criteria set out but he International Committee of Medical Journal Editors.

## Funding

B.B. is supported by a British Heart Foundation Clinical Research Fellowship (FS/CRTF/22/24437) and N.H. is supported by a British Heart Foundation Senior Clinical Research Fellowship (FS/SCRF/20/32005).

## Acknowledgements

Professors David Paterson and Keith Channon have been invaluable and important mentors over many years, and we are particularly grateful and indebted for their support. We are also extremely grateful to previous group members who have worked on NPY (in particular Julia Shanks, Chieh-Ju Lu, Manish Kalla, Nidi Tapoulal, Guoliang Hao, Harvey Davis, Tom Gibbs, Jakub Tomek and Peregrine Green), as well as our many collaborators, in particular the OxAMI investigators, and Professors Ajijola, Vaseghi, Ardell and Shivkumar at UCLA, Professor Habecker at Oregon Health Sciences, and Professors Jhund and McMurray at the University of Glasgow, who have been instrumental in many of the studies described here.

## Keywords

arrhythmia, autonomic nervous system, coronary microvascular function, heart failure, neuropeptide Y, sympathetic nervous system, ventricular fibrillation

## Supporting information

Additional supporting information can be found online in the Supporting Information section at the end of the HTML view of the article. Supporting information files available:

**Peer Review History**

