## [Peer Review History · The Journal of Physiology]

Bayliss-Starling Prize Lecture

Benjamin Mothibe Bussmann, Thamali Ayagama, Kun Liu, Dan Li, and Neil Herring
DOI: 10.1113/JP285370

Corresponding author(s): Neil Herring (neil.herring@dpag.ox.ac.uk)

The following individual(s) involved in review of this submission have agreed to reveal their identity: Corey Smith (Referee #1)

Review Timeline:	Submission Date:	05-Mar-2024
	Editorial Decision:	02-Apr-2024
	Revision Received:	19-Apr-2024
	Accepted:	01-May-2024

Senior Editor: Harold Schultz

Reviewing Editor: Kalyanam Shivkumar

Transaction Report:

Dear Professor Herring,

Re: JP-PL-2024-285370 "Bayliss-Starling Prize Lecture" by Benjamin Mothibe Bussmann, Thamali Ayagama, Kun Liu, Dan Li, and Neil Herring

Thank you for submitting your manuscript to The Journal of Physiology. It has been assessed by a Reviewing Editor and by 2 expert referees and we are pleased to tell you that it is acceptable for publication following satisfactory revision.

REVISION CHECKLIST:

We look forward to receiving your revised submission.

Yours sincerely,

Harold Schultz
Senior Editor
The Journal of Physiology

REQUIRED ITEMS:

- Please include an Abstract Figure file, as well as the Figure Legend text within the main article file. The Abstract Figure is a piece of artwork designed to give readers an immediate understanding of the Review Article and should summarise the main conclusions. If possible, the image should be easily 'readable' from left to right or top to bottom. It should show the physiological relevance of the Review so readers can assess the importance and content of the article. Abstract Figures should not merely recapitulate other figures in the Review. Please try to keep the diagram as simple as possible and without superfluous information that may distract from the main conclusion of the Review. Abstract Figures must be provided by authors no later than the revised manuscript stage and should be uploaded as a separate file during online submission labelled as File Type 'Abstract Figure'. Please ensure that you include the figure legend in the main article file. All Abstract Figures will be sent to a professional illustrator for redrawing and you may be asked to approve the redrawn figure before your paper is accepted.

- Your MS must include a complete "Additional information section" with the following 4 headings and content:

Competing Interests: A statement regarding competing interests. If there are no competing interests, a statement to this effect must be included. All authors should disclose any conflict of interest in accordance with journal policy.

Author contributions: Each author should take responsibility for a particular section of the study and have contributed to writing the paper. Acquisition of funding, administrative support or the collection of data alone does not justify authorship; these contributions to the study should be listed in the Acknowledgements. Additional information such as 'X and Y have contributed equally to this work' may be added as a footnote on the title page.

It must be stated that all authors approved the final version of the manuscript and that all persons designated as authors qualify for authorship, and all those who qualify for authorship are listed.

Funding: Authors must indicate all sources of funding, including grant numbers. If authors have not received funding, this must be stated.

It is the responsibility of authors funded by RCUK to adhere to their policy regarding funding sources and underlying research material. The policy requires funding information to be included within the acknowledgement section of a paper. Guidance on how to acknowledge funding information is provided by the Research Information Network. The policy also requires all research papers, if applicable, to include a statement on how any underlying research materials, such as data, samples or models, can be accessed. However, the policy does not require that the data must be made open. If there are considered to be good or compelling reasons to protect access to the data, for example commercial confidentiality or legitimate sensitivities around data derived from potentially identifiable human participants, these should be included in the statement.

Acknowledgements: Acknowledgements should be the minimum consistent with courtesy. The wording of acknowledgements of scientific assistance or advice must have been seen and approved by the persons concerned. This section should not include details of funding.

- Please upload separate high quality figure files via the submission form.

- Author profile(s) must be uploaded via the submission form. Authors should submit a short biography (no more than 100 words for one author or 150 words in total for two authors) and a portrait photograph of the two leading authors on the paper. These should be uploaded and clearly labelled together in a Word document with the revised version of the manuscript. Any standard image format for the photograph is acceptable, but the resolution should be at least 300 DPI and preferably more. A group photograph of all authors is also acceptable, providing the biography for the whole group does not exceed 150 words.

- It is the authors' responsibility to obtain any necessary permissions to reproduce previously published material and to list these within the main article file. For information, please see: https://jp.msubmit.net/cgi-bin/main.plex?form_type=display_requirements#permissions.

EDITOR COMMENTS

Reviewing Editor:

Please respond to the reviewers' comments.

Senior Editor:

Thank you for submission of your Prize Lecture article to the Journal of Physiology for consideration. The article has been reviewed by experts in the field and found to be acceptable for publication pending adequate revision to address minor concerns raised. Please address all comments from the external referees as well as addressing the list of requirements or publication in the journal for review articles.

If any of the figures were originally published or modified from a figure in another publication, please provide permission from the journal(s) to reproduce and a citation to the article in the figure legend.

Please provide a copy of the revised manuscript with changes highlighted and so named.

REFEREE COMMENTS

Referee #1:

In this manuscript, Bussmann et al provide a comprehensive consideration of co-transmitters on cardiac disease and progression. The material is provided in a manner based on primary data and clinical observation. The scope of the work is impressive, with description ranging from single cell excitability to large clinical trials; both with equal precision and expertise. Overall, this is a high value and high impact manuscript and the authors are congratulated on their achievement. This will certainly be highly cited for decades to come. With that level of excellence in mind, comments are minor and limited mainly to typographical errors.

There appears to be extra words at the end of line 31/start of line 32.

Lines 152-154 are redundant to lines 127-129. If the authors choose to alter this text, keeping the latter would provide better flow and context to the section on natriuretic peptides.

Typo in line 252

In line 306, NA is used for the first time in the text. It is identified in figure 1 legend but may be identified here as well.

Line 365 has a extra word, "stimulation".

Line 391, it seems that vagal nerve stimulation elevates fibrillation threshold, not reduces.

Line 442, "or" may be intended to be "of".

Line 447, is there a missing "and" between "with" and "all"?

Line 549, "by bass" should be "bypass"?

Again, congratulations to the authors on generating this fine manuscript.

Referee #2:

In this commissioned manuscript, the authors describe their work on sympathetic co-transmitter neuropeptide-Y (NPY) in cardiac disease. The manuscript begins with an apt initial description of the history of the Bayliss-Starling Prize Lecture. The lecture then delves in the role of the cardiac autonomic nervous system in cardiac physiology and pathophysiology with a focus on neuropeptide-Y (NPY). The lecture includes an overview of intrinsic neuromodulation, paracrine and endocrine modulation, and neural co-transmission with neuropeptides such as NPY, the signaling pathways of which are covered in detail. The loss of sympathovagal balance and neural remodeling in cardiac disease is reviewed. The prognostic implications of plasma NPY in patients with myocardial infarction and/or heart failure are discussed. These observations are then supported by mechanistic data determined in animal models and in vitro. The lecture concludes with diagnostic and therapeutic implications of the above findings. Overall, the lecture is well-written, thorough with supporting data spanning from in vitro models to human patients, and appropriately referenced. Dr. Herring is to be commended for his command of the literature and an impressive body of work. Such work is necessary to further our understanding of neural remodeling in cardiac disease and for the development of neuromodulation therapies. The figures are informative and easy to read. Figure 1 provides a helpful summary of the effects of NPY in the cardiac autonomic nervous system while Figures 2 and 3 display a composite of previously collected human data. I only have minor comments on typographical errors as follows:

1. Pg. 6, ln. 110: "its receptors" should read " its receptors' "
2. Pg. 7, ln 161: "endocine" should read "endocrine"
3. Pg. 9, ln. 252: "assicaited" should read "associated"
4. Pg. 9, ln. 253: "pepdites" should read "peptides"
5. Pg. 15, ln. 528: "Inhibition" should read "inhibition"
6. Pg. 15, ln. 537: "diagnositic" should read "diagnostic"
7. Pg. 17, ln. 626: "ability of" should read "ability to"
8. Fig.1: Font for adrenergic (beta 1?) receptor label should be corrected

END OF COMMENTS

Confidential Review

05-Mar-2024

EDITOR COMMENTS

Reviewing Editor:

Please respond to the reviewers' comments.

Senior Editor:

Thank you for submission of your Prize Lecture article to the Journal of Physiology for consideration. The article has been reviewed by experts in the field and found to be acceptable for publication pending adequate revision to address minor concerns raised. Please address all comments from the external referees as well as addressing the list of requirements or publication in the journal for review articles.

If any of the figures were originally published or modified from a figure in another publication, please provide permission from the journal(s) to reproduce and a citation to the article in the figure legend.

Response: Figures have been modified from publications which are open access article distributed under the terms of the CC BY 4.0 license, which permits unrestricted use, distribution, and reproduction in any medium, provided the original work is properly cited. We are not required to obtain permission to reuse.

Please provide a copy of the revised manuscript with changes highlighted and so named.

Response: We have provided a copy of the manuscript with changes highlighted

REFEREE COMMENTS

Referee #1:

In this manuscript, Bussmann et al provide a comprehensive consideration of co-transmitters on cardiac disease and progression. The material is provided in a manner based on primary data and clinical observation. The scope of the work is impressive, with description ranging from single cell excitability to large clinical trials; both with equal precision and expertise. Overall, this is a high value and high impact manuscript and the authors are congratulated on their achievement. This will certainly be highly cited for decades to come. With that level of excellence in mind, comments are minor and limited mainly to typographical errors.

There appears to be extra words at the end of line 31/start of line 32.

Response. Now corrected. Thank you

Lines 152-154 are redundant to lines 127-129. If the authors choose to alter this text, keeping the latter would provide better flow and context to the section on natriuretic peptides.

Response: We have changed the wording to remove redundancy but still highlight that both nNOS and natriuretic peptides augment ACh release through a similar cGMP-PDE dependant pathway

Typo in line 252

Response: Now corrected. Thank you.

In line 306, NA is used for the first time in the text. It is identified in figure 1 legend but may be identified here as well.

Response: We have replaced NA with noradrenaline to keep this consistent throughout the manuscript.

Line 365 has a extra word, "stimulation".

Response: Now corrected. Thank you

Line 391, it seems that vagal nerve stimulation elevates fibrillation threshold, not reduces.

Response: The reviewer is correct. Now amended. Thank you

Line 442, "or" may be intended to be "of".

Response: Thank you. Now corrected.

Line 447, is there a missing "and" between "with" and "all"?

Response: Thank you, we have corrected this

Line 549, "by bass" should be "bypass"?

Response: Now corrected. Thank you

Again, congratulations to the authors on generating this fine manuscript.

Referee #2:

In this commissioned manuscript, the authors describe their work on sympathetic co-transmitter neuropeptide-Y (NPY) in cardiac disease. The manuscript begins with an apt initial description of the history of the Bayliss-Starling Prize Lecture. The lecture then delves in the role of the cardiac autonomic nervous system in cardiac physiology and pathophysiology with a focus on neuropeptide-Y (NPY). The lecture includes an overview of intrinsic neuromodulation, paracrine and endocrine modulation, and neural co-transmission with neuropeptides such as NPY, the

signaling pathways of which are covered in detail. The loss of sympathovagal balance and neural remodeling in cardiac disease is reviewed. The prognostic implications of plasma NPY in patients with myocardial infarction and/or heart failure are discussed. These observations are then supported by mechanistic data determined in animal models and in vitro. The lecture concludes with diagnostic and therapeutic implications of the above findings. Overall, the lecture is well-written, thorough with supporting data spanning from in vitro models to human patients, and appropriately referenced. Dr. Herring is to be commended for his command of the literature and an impressive body of work. Such work is necessary to further our understanding of neural remodeling in cardiac disease and for the development of neuromodulation therapies. The figures are informative and easy to read. Figure 1 provides a helpful summary of the effects of NPY in the cardiac autonomic nervous system while Figures 2 and 3 display a composite of previously collected human data. I only have minor comments on typographical errors as follows:

1. Pg. 6, ln. 110: "its receptors" should read " its receptors' "

Response: Now corrected. Thank you

2. Pg. 7, ln 161: "endocrine" should read "endocrine"

Response: Now corrected. Thank you

3. Pg. 9, ln. 252: "assicated" should read "associated"

Response: Now corrected. Thank you]

4. Pg. 9, ln. 253: "pepdites" should read "peptides"

Response: Now corrected. Thank you

5. Pg. 15, ln. 528: "Inhibition" should read "inhibition"

Response: Thank you

6. Pg. 15, ln. 537: "diagnositc" should read "diagnostic"

Response: Thank you

7. Pg. 17, ln. 626: "ability of" should read "ability to"

Response: We agree. Thank you

8. Fig.1: Font for adrenergic (beta 1?) receptor label should be corrected

Response: Now corrected. Thank you

END OF COMMENTS

Dear Professor Herring,

Re: JP-PL-2024-285370R1 "Bayliss-Starling Prize Lecture" by Benjamin Mothibe Bussmann, Thamali Ayagama, Kun Liu, Dan Li, and Neil Herring

We are pleased to tell you that your paper has been accepted for publication in The Journal of Physiology.

Authors should note that it is too late at this point to offer corrections prior to proofing. The accepted version will be published online, ahead of the copy edited and typeset version being made available. Major corrections at proof stage, such as changes to figures, will be referred to the Editors for approval before they can be incorporated. Only minor changes, such as to style and consistency, should be made at proof stage. Changes that need to be made after proof stage will usually require a formal correction notice.

All queries at proof stage should be sent to: TJP@wiley.com

Yours sincerely,

Harold Schultz
Senior Editor
The Journal of Physiology

P.S. - You can help your research get the attention it deserves! Check out Wiley's free Promotion Guide for best-practice recommendations for promoting your work at www.wileyauthors.com/eeo/guide. You can learn more about Wiley Editing Services which offers professional video, design, and writing services to create shareable video abstracts, infographics, conference posters, lay summaries, and research news stories for your research at www.wileyauthors.com/eeo/promotion.

IMPORTANT NOTICE ABOUT OPEN ACCESS: To assist authors whose funding agencies mandate public access to published research findings sooner than 12 months after publication The Journal of Physiology allows authors to pay an Open Access (OA) fee to have their papers made freely available immediately on publication.

You can check if your funder or institution has a Wiley Open Access Account here: <https://authorservices.wiley.com/author-resources/Journal-Authors/licensing-and-open-access/open-access/author-compliance-tool.html>

Senior Editor Comments:

The editors wish to thank the authors for their adjustments to the manuscript. The article is now accepted for publication. Congratulations for an interesting and insightful review article. Please consider the Journal of Physiology for your future studies.

Reviewing Editor Comments:

No further comments.

Referee 1 Comments:

Excellent manuscript. No further concerns or comments.

Referee 2 Comments:

My comments have been satisfactorily addressed. Congratulations again on an impressive body of work.

1st Confidential Review

19-Apr-2024